# School Health Education and Teachers' Preservice Training: The Case of Greece

Pelagia Soultatou [1],* and Kyriacos Athanasiou [2]

1   Department of Education Sciences, School of Humanities, Social & Education Sciences,
    European University Cyprus, 2404 Nicosia, Cyprus
2   Department of Early Childhood Education, School of Education Sciences,
    National and Kapodistrian University of Athens, 10680 Athens, Greece; kathanas@ecd.uoa.gr
*   Correspondence: p.soultatou@external.euc.ac.cy

**Abstract:** Background: School health education (SHE) serves as a primary pedagogical aspect of public health. This study aims to explore the curricula of preservice schoolteachers, examining whether graduate courses in Greek universities include SHE, how it is structured, and whether critical pedagogy principles are integrated. Methods: A corpus of $n = 21$ documents was compiled, encompassing all educational departments in Greek higher education, including Departments of Early Childhood Education and Care ($n = 3$), Departments of Preschool Education ($n = 9$), and Departments of Primary Education ($n = 9$). Results: Content analysis revealed the presence of SHE in 11 out of 21 educational departments. Thematic analysis uncovered significant variability in the curricula, with greater emphasis placed on health-related topics rather than the methodologies and values of SHE. Critical pedagogy principles were not explicitly evident in the curricula, while a persistent biomedical orientation left little room for the development of concepts related to social justice. Conclusions: This study represents the first of its kind in Greece and among the few internationally to examine preservice training for schoolteachers in SHE. The findings underscore the need for revisions to align with the key lessons learned from the major public health crisis of the COVID-19 pandemic. It is imperative to acknowledge that vulnerabilities have been exacerbated, and inequalities widened, necessitating a reassessment of current approaches to health education within teacher training programs.

**Keywords:** health education; curriculum; teachers; university; preservice training

## 1. Introduction

School, as a social apparatus, serves not only as a center for learning but also as a platform for fostering social awareness, shaping social consciousness, and promoting solidarity within the school community as a whole [1]. This expanded view of schooling extends beyond the formal curriculum to encompass the informal and hidden curricula. The formal curriculum consists of officially designed and mandated activities that are allocated specific teaching periods within the national timetable and are compulsory in nature. In contrast, the informal curriculum comprises voluntary and non-compulsory activities that accompany educational practice, while the hidden curriculum encompasses the implicit messages embedded within the prevailing school culture, including classroom materials, cultural artifacts, and rituals within the schooling process [2].

Although school health education may not be explicitly incorporated into the formal curriculum, it may remain inherent within the informal and hidden curricula. These aspects are integral to what children learn through the prominent cultural features of the school environment, perpetuating and reinforcing the norms and values conveyed in the formal curriculum. The primary agent responsible for embodying the hidden curriculum is the schoolteacher, who provides avenues for social and emotional learning and well-being through various aspects of the schooling process, such as informal instruction, role

modeling, and mentoring. These practices are intricately linked to the evolving narrative ecology of schools, particularly in the context of public health crises [3]. However, it is acknowledged that preservice teacher education alone is insufficient to fully shape the professional identity of educators. Professional identity formation is a dynamic process that results from the ongoing interplay between preservice teacher education, vocational integration, the execution of professional duties, and in-service training [4].

In the wake of the unprecedented public health crisis brought about by the COVID-19 pandemic, a crucial question emerges: How will future generations be educated on public health issues within the schooling process? To critically address this broad yet compelling question, it is imperative to first examine the preservice training of schoolteachers, which leads us to inquire: Do future schoolteachers receive adequate preservice training in health-related topics and methods, and if so, what form does this training take? This study focuses on the Greek higher education system, identifying the faculties within public universities that provide preservice training to future schoolteachers and investigating whether health education is included in their syllabi, aiming to map out the curricula concerning the content and methods of health education.

## 2. Theoretical Background

### 2.1. Health Education: Significance and Evolution

Health education has been a recognized field of knowledge and action for over two centuries, and yet it often remains marginalized within the broader public health domain and educational discourse. Its origins can be traced back to the late eighteenth and early nineteenth centuries in industrialized countries. During this period, the migration from rural to urban areas, coupled with overcrowding, poverty, and inadequate sanitation, led to a surge in epidemic diseases, necessitating public health interventions and educational efforts to inform the populace about health-related matters [5]. Throughout its history, health education has taken various forms, ranging from authoritarian approaches during times of crisis to more emancipatory practices that emphasize solidarity and view health as a public good, in line with the mandates of the World Health Organization for equity [6–9]. While the conceptualization of health education varies widely in the literature, this article views it as the primary pedagogical method of public health. However, it is crucial to acknowledge that health education is a value-laden and contested domain, imbued with political considerations and conceptualized within the epistemological dichotomy between health objectivism and health interpretivism.

### 2.2. Health Objectivism versus Health Subjectivism

Health objectivism, rooted in the biomedical perspective and championed by Parsonian scholars in sociology, posits that health is an objective fact that can be empirically observed. This perspective often simplifies health as merely the absence of disease, with little regard for social and environmental factors [10,11]. In contrast, health subjectivism challenges this dominant paradigm by asserting that meanings related to health are constructed, negotiated, and managed by individuals and groups within various social and historical contexts. It emphasizes democratic education, critical thinking, action, and a holistic view of health, incorporating environmental considerations [12].

To address the persistent regression of health education towards objectivism and individualism, many scholars advocate for revisiting and revising health education curricula with principles of critical pedagogy. This approach emphasizes the importance of questioning power dynamics, promoting critical thinking, and fostering a more holistic understanding of health [6–9,13,14]. Given that social inequalities have been exacerbated during the health crisis of the pandemic [15,16], empowerment education becomes crucial to ensure community participation and dialogue at both personal and social levels. This approach aims to strengthen individuals' and communities' ability to gain control over their own health. Practical strategies derived from the literature advocate for the implementation of a three-phase model of critical pedagogy in health education. This model addresses

inherent difficulties in using this teaching approach, such as fostering a power-sharing arrangement between students and teachers, negotiating learning content, ensuring the inclusivity of diverse voices and perspectives, and navigating potential conflicts arising from the politically charged nature of the subject matter.

*2.3. Health Education in the Post-Pandemic Era*

The COVID-19 pandemic has challenged the dominance of the biomedical model on two fronts. Firstly, it has demonstrated that the threat of massive infectious diseases persists despite biomedical advancements. Rather than disappearing, epidemics are likely to recur due to underlying causes, including the transformation of agricultural practices and habitat destruction [17]. Secondly, the pandemic has exposed a deep-seated resistance to biomedical hegemony. The proliferation of misinformation, termed an "infodemic", has fueled vaccine hesitancy and prompted governments worldwide to adopt a militaristic rhetoric. Mass communication channels have perpetuated martial and patriarchal metaphors, which are inherently masculine, power-based, paternalistic, and violent [18]. This infodemic can be seen as both a challenge to medical dominance and a rejection of health objectivism. The pervasive use of war language during emergencies should be replaced with health education methods grounded in egalitarian and democratic principles to combat vaccine hesitancy and misinformation [19–21]. The massive public health crisis was also accompanied by the notion of solidarity as a means to cultivate alliances against the common "enemy", as the COVID-19 has been viewed. Adopting therefore a critical public health and critical pedagogy perspective, and the principles of equality, equity, democracy, inclusion and grassroots participation decision, will flourish in a radical ecological model [7,22] against the de-politicization of public health. At the heart of this theoretical model lies empowerment education, which ensures community participation and dialogue at a personal level but also in social arenas to reinforce the individuals' and communities' ability to gain control over their own health [7].

*2.4. School Health Education in Europe*

School health education is allied with primary healthcare services provided in schools by health professionals and has the potential to address health inequalities for vulnerable children in deprived regions and lower socioeconomic strata but also to offer health services and effective interventions, fulfilling the goals of disease prevention and health promotion. However, a recent large-scale comparative study based on data from 30 European countries documented substantial disparities between countries regarding health services and reported deficient training in school health education [23].

The teaching of health literacy serves as a principal component of the whole-school approach, such as the WHO's Health Promoting Schools framework. A scoping review was conducted in the WHO European Region exploring the parameters of 46 health literacy in education policies at the national, regional, or local level, seeking to integrate measures and actions related to organizational change, workforce development, intersectoral partnerships, and education policy [24]. One of the key conclusions of this particular report by the WHO is the substantial disparities amongst European countries and the need to develop and implement holistic health literacy policies in member states and a rigorous evaluation of policy-related activities to demonstrate the benefits of health literacy policies to citizens and society.

**3. Materials and Methods**

This empirical work aims to investigate the following: (a) whether health education is included as a subject in undergraduate courses offered by educational departments at public universities in Greece; (b) how health education is structured within the curriculum of each educational department; and (c) to what extent critical pedagogy principles are integrated into the subject of health education.

The study focuses on the following educational departments: (a) Departments of Early Childhood Education and Care; (b) Departments of Early Childhood Education; and (c) Departments of Primary Education.

The research questions guiding the data analyses are as follows:

1. Is "health education" (HE) included as a subject in the undergraduate courses of Departments of Education at public Greek universities?
2. In what form (mandatory or optional) is the subject of HE included in these undergraduate courses?
3. How is HE described in the curriculum of each Department of Education?
4. Does the HE curriculum follow the paradigm of health objectivism or health subjectivism?
5. Does the HE curriculum address methodological and ethical issues in health education?
6. To what extent can critical pedagogy principles be identified in the syllabuses of HE modules within the graduate courses offered by these Departments of Education?

Content analysis and thematic analysis have been employed as analytical approaches for the body of textual data derived from higher education curricula. Initially, content analysis was utilized manually to systematically identify and quantify the presence of health education in the curricula of educational departments, thus addressing the first two research questions. The codes generated for this segment of data were "health education" and "health literacy" through a corpus of seventeen undergraduate studies syllabuses, encompassing both preschool and primary education. Inclusion criteria include terms such as "health" and "education" and/or "literacy" as well as terms related to health promotion and prevention. Exclusion criteria encompassed codes such as "special needs", "mental health", "physical education", "first aids", and "biology". In the second round of analyses, we sought to locate the terms "solidarity", "critical", "collaboration" and/or "alliances", "partnerships", "values", "democracy", "empowerment", and "equality" that correspond to the principles of critical pedagogy.

Subsequently, content analysis was used to explore the content of each curriculum in response to research questions three to six. The data were analyzed to identify recurring themes and patterns regarding health-related topics, pedagogical principles, and critical pedagogy concepts within the curricula.

## 4. Results

The study compiled a corpus of 21 documents, corresponding to the total number of educational departments in the Greek higher education system. These documents represent three segments of educational departments: Departments of Early Childhood Education and Care ($n = 3$), Departments of Early Childhood Education ($n = 9$), and Departments of Primary Education ($n = 9$).

Overall, HE is identified in 10 out of 21 educational departments (47%). Specifically, HE is included in 66% of Departments of Early Childhood Education and Care, 44% of Departments of Preschool Education, and 44% of Departments of Primary Education. Tables 1–3 summarize the findings from each segment of educational departments. Notably, the curriculum analysis reveals variations in the inclusion of health-related topics, pedagogical methods, and critical pedagogy principles across different departments.

Table 1 presents data from Departments of Early Childhood Education and Care that include HE in their syllabuses. For instance, the University of West Attica emphasizes biomedical-oriented health topics, while the University of Ioannina allocates space for pedagogical principles but lacks explicit references to critical pedagogy.

Table 2 above presents data from Departments of Early Childhood Education, where a portion of departments integrates HE into their syllabuses. Notably, the National and Kapodistrian University of Athens and the Democritus University of Thrace incorporate both health-related topics and pedagogical methods, with indications of critical pedagogy principles.

The curriculum of the National and Kapodistrian University of Athens is structured into two distinct sections. The first section focuses on core health-related topics to be

covered throughout the academic semester, while the second section addresses pedagogical methods inherent to HE, allowing space for critical pedagogy principles while maintaining a balance between topics and methods.

Similarly, the curriculum of the Democritus University of Thrace also distinguishes between content and methods, highlighting the introduction of a "modern methodology" of health education. This approach suggests a departure from traditional methods, emphasizing the development of skills that promote mental, physical, and social well-being.

In contrast, the curriculum of the University of Western Macedonia is structured around 12 modules, with only 3 modules focusing on health-related topics such as nutrition, oral hygiene, and traffic education. These topics do not exhibit a strong biomedical orientation. Instead, most of the modules (9 out of 12) focus on methodological and conceptual/ethical dimensions of HE, leaving ample space for the integration of critical pedagogy principles.

Overall, the analysis of curricula across different departments highlights variations in the incorporation of health-related topics, pedagogical methods, and critical pedagogy principles. While some departments demonstrate a balanced approach between content and methods, others prioritize methodological and conceptual dimensions of HE, with varying degrees of emphasis on critical pedagogy principles.

**Table 1.** Departments of Early Childhood Education and Care.

| Institution | Subject | Status | Content |
|---|---|---|---|
| University of West Attica (Aigaleo, Greece) | Health Education | Mandatory 2 h pw 3 ECTS | Public health principles. Hygiene in education and care in early childhood. Safety in education and care areas in early childhood. Principles of healthy eating in early childhood. Nutrition and physical activity in early childhood. Food hygiene in education and care areas. Principles of prevention of communicable and non-communicable diseases in early childhood. First aid—the first steps of first aid—The contents of a pharmacy for first aid in the nursery—Medicines in education and care areas. Injuries from mechanical causes. Injuries from natural causes. Foreign bodies (swallowing, etc.). Poisoning and antidotes. Pathological causes that need first aid. Pulse, pain. Introduction to artificial respiration and cardiorespiratory resuscitation (CPR). |
| International Hellenic University (Thermi, Greece) | | | |
| University of Ioannina (Ioannina, Greece) | Health Education | Mandatory 3 h pw 4 ECTS | Definitions of health and health education. Historical overview and mythology. Nutrition and health. Cardiovascular diseases. Prevention of cancer. Smoking. Mental health and emotional management. Prevention of the use of addictive substances. Traffic and accident prevention. Environmental impacts on health. Genetic diseases and sexual education. Teeth and health. Emergency treatment. |

**Table 2.** Departments of Early Childhood Education.

| Institution | Subject | Status | Content |
|---|---|---|---|
| National and Kapodistrian University of Athens (Athens, Greece) | Health Education | Optional 3 h pw 5 ECTS | The first part includes themes such as nutrition, dental care, cancer prevention, the prevention of cardiovascular and genetic diseases, and information on sex education issues. Because environmental concerns and their effects on health tend to occupy an important part of health topics nowadays, the course material also includes chapters on the risks associated with smoking, other environmental pollutants (lead and asbestos), and exposure to radiation. Finally, because health education seems to be creating its own teaching methodology over time, it was deemed necessary to incorporate modules into the course material for modern teaching methods related to sex education and nutrition. |
| Democritus University of Thrace (Komotini, Greece) | Health Education for preschool age | Optional 3 h pw 4.5 ECTS | Detailed mental health promotion programs are presented, such as programs to boost self-esteem and self-confidence, develop communication skills, resolve conflicts, and manage emotions. There are programs of nutrition education, physical exercise and hygiene, protection from accidents and natural disasters, programs that aim at healthy interracial relationships in the future but also at the avoidance of addictions. Students are introduced to the modern methodology of health education, which concerns the development of those skills that promote mental and physical health, but also social well-being. |
| University of Western Macedonia (Kozani, Greece) | Health Education | Optional 3 h pw 4 ECTS | Definitions of health and illness, health education, and health promotion. The factors that affect health behavior. Basic theories of health education and models of interventions. Children's perceptions of the human body, health, and disease. Nutrition issues. Oral hygiene issues. Traffic education issues. Consumer education issues. Health education in the Hellenic educational system. Educational materials and selection criteria. Designing school health education program promotions for young children. Collaborating with parents and the community. |

The curriculum of the University of Ioannina offers two HE subjects in the syllabus: one mandatory (Health Education I) and one optional (Health Education II). However, the syllabus lacks a description of the curriculum for the optional subject (i.e., Health Education II). Regarding HE I, 11 out of 13 modules (from the 3rd to the 13th) are topic-oriented, while only 2 out of 13 (i.e., the 1st and 2nd) leave space for pedagogical principles and critical pedagogy.

Table 3 above presents data extracted from all the Departments of Primary Education in Greek public universities. The graduates will be involved in primary education for children aged 6 to 11 years old.

As depicted in Table 3, four out of nine departments of primary education include HE as a subject in their syllabuses, covering a total of 44%. Notably, at Aristotle University,

while the subject is included in the timetable of the Department, it is not incorporated into the syllabus, resulting in the absence of a curriculum.

**Table 3.** Departments of Primary Education.

| Institution | Subject | Status | Content |
|---|---|---|---|
| National and Kapodistrian University of Athens | Health Education | Optional 4 ECTS (hours not specified) | Conceptual definition, goals and methods of health education. Schools that promote health and the role of the teacher in the school. Health education. Theoretical approach and didactic applications [selection, implementation and evaluation of work plans (projects)—role-playing games—public debates (debates, etc.)] in the thematic units: self-esteem, self-confidence, nutrition, physical exercise, use and abuse of substances, sexual treatment, oral health treatment, and stress. |
| Aristotle University of Thessaloniki (Thessaloniki, Greece) | Health Education in Primary Education | Optional 4 ECTS (hours not specified) | Not described |
| University of Ioannina (Ioannina, Greece) | Health literacy and experimental methods | Optional 5 ECTS 3 h pw | Application of knowledge in practice. Work in an interdisciplinary environment. Decision making. Respect for diversity and multiculturalism. Adaptation to new situations. Respect for the natural environment. Autonomous work. Group work. Generating new research ideas. Project planning and management. Movement and knowledge about life. |
| University of Patras (Patras, Greece) | Health Education | Optional 5 ECTS 3 h pw | Unit A: lessons 1–2: Health, health education—Health promotion, health literacy. Section B: lessons 3–6: Programs—Project, logical model, planning—Educational activities. |
| Aristotle University of Thessaloniki (Thessaloniki, Greece) | | | |
| University of the Aegean (Mytilene, Greece) | | | |
| Democritus University of Thrace (Komotini, Greece) | | | |
| University of Crete (Rethymno, Greece) | | | |
| University of Thessaly (Volos, Greece) | | | |

In terms of content and methods, the National and Kapodistrian University offers two subjects, with one primarily focused on HE. The curriculum includes conceptual approaches and methods of HE following a school setting approach. The topics cover behavior issues (e.g., self-esteem and self-confidence) and elementary individual health (e.g., physical exercise, substance use and abuse, sexual health, oral health, and stress).

At the University of Ioannina, the curriculum is primarily oriented towards methods (e.g., the application of knowledge in practice, interdisciplinary work, decision making, teamwork, and project planning) rather than health-related topics, providing ample space for critical pedagogy principles to flourish.

Similarly, the curriculum at the University of Patras strikes a balance between fundamental concepts of HE (e.g., health, health education, health promotion, and health literacy) and methods (e.g., programs, project management, and planning educational activities).

Overall, across Departments of Primary Education, there is variation in the inclusion of HE, with some departments prioritizing topics and methods while others maintain a balance between the two, allowing for the integration of critical pedagogy principles.

## 5. Discussion

This study represents a pioneering effort within the Greek higher education system, addressing the training of preservice schoolteachers in school health education (SHE), particularly in the aftermath of the COVID-19 pandemic. It aims to ascertain whether SHE is incorporated as a subject in preservice teacher training and to examine the structure of SHE curricula within educational departments. In respect to the first research question, concerning the inclusion of SHE as a subject in the undergraduate courses of Departments of Education at public Greek universities, it was revealed that SHE was included in only 11 out of the total 21 educational departments, which corresponds to a 47% presence of SHE in preservice teachers' training. The character of the subject was mostly that of an optional module. These findings raise concerns about the marginal provision of preservice SHE training to future schoolteachers in both preschool and primary education. Given the heightened awareness of the importance of SHE in managing public health crises, such as the COVID-19 pandemic, the minimal presence of SHE in teaching bachelor's degrees may be interpreted as indicative of a bias towards biomedical approaches, potentially undermining the recognition of SHE as an essential component of education and associating it primarily with medical, nursing, and public health schools rather than educational faculties.

While the literature on this topic is scarce, previous studies have touched upon related issues. For instance, a study examining bachelor's degree courses in primary, secondary, and physical education at Canadian universities found that preservice sexual health education training was offered below the average, with only 39.3% of courses including it either as a compulsory or optional subject [25]. However, this study focused specifically on sexual health education. Another study, focusing on curriculum changes in health promotion provision in preservice teacher education in a postgraduate certificate course in education secondary program at a UK Higher Education Institution, explored transformations to the health promotion component of the program and their implementation [26].

The findings of the current study underscore the need for greater attention to SHE within preservice teacher training programs, particularly in the context of ongoing and future public health crises. Efforts should be made to integrate SHE more comprehensively into educational curricula, emphasizing its importance in promoting a holistic well-being and addressing societal health challenges. This could involve revising existing curricula, providing specialized training for educators, and fostering collaboration between educational and public health institutions to ensure that preservice teachers are equipped with the necessary knowledge and skills to effectively address health-related issues in school settings. This study sheds light on a critical aspect of preservice teacher education and underscores the importance of prioritizing SHE within educational curricula. By addressing the gaps identified in this study, policymakers and educators can better prepare future generations of schoolteachers to promote health and well-being among students, contributing to the overall improvement of public health outcomes.

The second aim of this article is to identify patterns within the SHE curriculum, focusing on understanding how the curriculum is shaped and the extent to which it encompasses health-related topics, methods, and ethics. Of particular importance within this empirical work is the tracking of critical pedagogy principles within the curriculum. The second aim of this study is framed by the following three research questions, namely, whether the SHE curriculum follows the paradigm of health objectivism or health subjectivism, whether it addresses methodological and ethical issues, and to what degree it is informed by critical pedagogy principles.

The findings indicate that graduates' preservice training in SHE tends to maintain an explicit biomedical orientation towards health-related topics. However, methods and ethics, which are foundational aspects of the subject, are often overlooked in the curriculum

design. Similarly, analyses of preservice primary education teachers' training reveal a similar trend, with some instances of the curriculum incorporating more sophisticated critical health promotion concepts (e.g., empowerment ande environmental respect) and methods (teamwork, decision-making skills, and project planning). This variability in the modus operandi of the curricula echoes previous evidence, highlighting inconsistencies in preservice teacher training approaches [25].

These concerns regarding the preservice training of schoolteachers in Greece are echoed in the existing literature. For example, an article emphasizing the need to integrate the teaching of science and biology with health and environmental education presents a proposal for a university course tailored to the needs of present and future teachers in these subjects [27]. Furthermore, a recent scoping review conducted in Greece underscores the necessity of revising SHE curricula with critical pedagogy principles. The review identifies ongoing challenges such as deficiencies in schoolteachers' in-service training, coordination issues, and leadership impediments that hinder the effective implementation of health education in schools [28].

These findings highlight the urgent need for reforms in preservice teacher education programs, particularly in the context of health education. Integrating critical pedagogy principles into SHE curricula can enhance the capacity of future schoolteachers to address complex health-related issues, promote critical thinking, and empower students to become active participants in their own well-being. Additionally, efforts to improve coordination, leadership, and ongoing professional development for schoolteachers are essential to ensure the successful implementation of health education initiatives in schools.

In conclusion, addressing the shortcomings identified in preservice teacher training programs is crucial for advancing health education in Greek schools and empowering future generations to make informed decisions about their health and well-being. By incorporating critical pedagogy principles and fostering collaboration between educational institutions, policymakers, and public health authorities, Greece can take significant strides towards promoting a culture of health and equity within its school systems.

Health education has undergone a significant evolution, shifting from a focus on behavior change to broader goals of creating healthy, fair, and sustainable environments, with an emphasis on social justice. In the wake of the COVID-19 pandemic, there is a renewed urgency to rejuvenate the discourse and practice of health promotion, with a focus on planetary health issues, civil society engagement in policymaking, and promoting health equity [29].

However, the existing curriculum design in many educational departments in Greece lacks explicit elements of critical pedagogy principles. Integrating critical pedagogy with health education can foster a co-learning and power-sharing environment, promote a democratic and inclusive atmosphere, and facilitate conflict resolution towards emancipation and social transformation [14,30]. Policymakers and tertiary education bodies should prioritize the inclusion of SHE as a subject in preservice teacher training programs, while also enhancing the content and methods incorporated into higher education curricula to advance the values of solidarity inherent in critical pedagogy.

However, challenges remain, as previous scholarly work has highlighted the dominance of general pedagogical knowledge and traditional didactic approaches in many educational departments in Greece. The COVID-19 pandemic further underscored the need to rethink teaching methodologies, with the forced migration to digital learning environments. While digital technologies offer opportunities for innovative teaching methods, there is a need to ensure that SHE methods and values are integrated effectively into digital learning platforms to effectively address public health crises.

In conclusion, addressing these challenges requires a concerted effort from policymakers, educators, and stakeholders to prioritize health education, integrate critical pedagogy principles, and leverage digital technologies to enhance teaching and learning experiences. By doing so, Greece can make significant strides towards promoting health equity, social justice, and a holistic well-being among its citizens.

## 6. Conclusions

In a broader perspective, the recent massive public health and humanitarian crisis has underscored the importance of educating populations not only in individual health protection measures but also in cultivating a culture of solidarity to address social inequalities and achieve social justice. It is imperative to move away from a health objectivist perspective towards health subjectivism, acknowledging the importance of egalitarian values and methods in educating children for potential future crises.

Higher education institutions offering preservice training to schoolteachers play a crucial role in addressing these challenges. Firstly, they must recognize the urgent need to reinforce health education within their curricula. This entails integrating health education as a subject and ensuring that it receives adequate attention in teacher training programs. Secondly, there is a need to revise existing curricula by incorporating critical pedagogy principles. These principles emphasize a transformative approach to education, promoting critical thinking, empowerment, and social justice. By incorporating critical pedagogy principles, teacher training programs can equip future educators with the tools and mindset needed to effectively address complex health and social issues.

Overall, aligning preservice teacher education with the principles of health subjectivism and critical pedagogy is essential for preparing educators to navigate the challenges of future public health crises and contribute to building a more equitable and just society. It requires a concerted effort from educational institutions, policymakers, and stakeholders to prioritize these principles and ensure their integration into teacher training programs.

**Author Contributions:** Conceptualization, P.S. and K.A.; Methodology, P.S.; Writing—Original Draft Preparation, P.S.; Writing—Review & Editing, P.S.; Supervision, K.A. All authors have read and agreed to the published version of the manuscript.

**Funding:** This research received no external funding.

**Institutional Review Board Statement:** Not applicable.

**Informed Consent Statement:** Not applicable.

**Data Availability Statement:** Data are contained within the article.

**Conflicts of Interest:** The authors declare no conflicts of interest.

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
