# Peer review of "School Health Education and Teachers’ Preservice Training: The Case of Greece"

_education, doi:10.3390/educsci14050483_

Round 1

Reviewer 1 Report

Comments and Suggestions for Authors

Dear Authors,

your manuscript evaluates the presence of healthy education in teachers' training, using content analyses of the curriculum of universities from Grace. 

I have some fundamental questions and I will present some gaps in your analyses:

- how is present healthy education in other higher education curriculums, and programs for teachers across Europe? What is the situation in other countries? The manuscript is very poor in this area. 

- you mention the principle of critical pedagogy, but is not clear, how this topic has been included in publications, on healthy education topics till now. 

- how can teachers teach this topic to children?

- how can this topic be included in all programs for students, avoiding overloads?

- is forbidden to use % for 21 documents!!!! 

- what softer did you use for content analyses?

- what codes did you use for content analyses? what categories and subcategories did you make? 

- in conclusion, must give a response to each of your research questions.   

Author Response

Dear Referee, 

thank you for affording time and effort to provide your valuable review that provoked the amendment of our manuscript. 

We reply to each of your comments:

  • we added a paragraph in the Introduction seeking to present in brief an overview of the health education in Europe
  • we added a paragraph in the Introduction seeking to connect more closely and more clearly the critical pedagogy with health education especially in the post pandemic era
  • the interesting question "how can teachers this topic to children" is beyond the scope of our work in this manuscript as we focus strictly to preservice teachers' training and not in-service teaching 
  • the percentage used has been now clarified
  • we did not use software for our analyses, hence we added that the coding was conducted manually
  • we provided further clarifications as for the coding
  • we connected further our research questions with the Discussion section.

However the question "how can this topic be included in all programs for students, avoiding overloads" was not understood.

Hopefully we utilised the feedback provided.

Reviewer 2 Report

Comments and Suggestions for Authors

Dear authors,

The study presented describes the education on health subjects dureing their studies in Greece.

The overall manuscript is well developed,although there are some areas of imrpovement:

1. The referencing could be improved, with areas of information with no reference o as I understood, using the reference at the end or in the next paragraph. In this sense, there are several references over 15 years old.

2. The study has focused on the existence or not of health related studies but I would have appreciated if the information was completed with the hours of training or ECTS assigned to each subject. This would provide a more in-depth information.

The fact that the syllabus of each university is described provides clarity and provides and added value.

Author Response

Dear Referee, 

thank you for affording time and effort to provide us with your valuable review that provoked the amendment of our manuscript. 

We reply to each one of your useful comments:

  • we added further information in the Tables I, II, III in respect to the status of the Health Education as a subject in the curricula of the educational departments by adding the ECTS and the hours assigned to the subject
  • we do understand the validity of your comment in respect to some outdated references, but we wish to clarify that most of them are classic theoretical texts and not current evidence. For instance we chose to maintain the reference of Giroux (2011) as one of the prominent figures in critical pedagogy, Kelly (2009) as it is a theoretical boon in curriculum studies and Baggott (2000), again as a classic textbook in public health domain. The reference Simpson & Freeman (2004) is maintained to show that history of the linkage between critical pedagogy with health education. Likewise Duncan' s (2004 and 2007) work is considered as philosophy of health which is timeless. Finally McKay and Barrett (1999) is used on purpose to show how scarce is the evidence on the topic we are focusing on.

Hopefully we utilised the feedback provided.